# Perspective of Molecular Diagnosis in Healthcare: From Barcode to Pattern Recognition

**DOI:** 10.3390/diagnostics9030075

**Published:** 2019-07-13

**Authors:** Qian He, Mengdi Bao, Kenneth Hass, Wenxia Lin, Peiwu Qin, Ke Du

**Affiliations:** 1Department of Mechanical Engineering, Rochester Institute of Technology, Rochester, NY 14623, USA; 2Precision Medicine and Public Healthcare Research Center, Tsinghua-Berkeley Shenzhen Institute, Shenzhen 518057, China; 3Department of QB3, University of California, Berkeley, CA 94720, USA; 4Department of Microsystems Engineering, Rochester Institute of Technology, NY 14623, USA

**Keywords:** bioassay, barcode, fluorescence, DNA, RNA, gold nanoparticle, point-of-care (POC)

## Abstract

Barcode technology has a broad spectrum of applications including healthcare, food security, and environmental monitoring, due to its ability to encode large amounts of information. With the rapid development of modern molecular research, barcodes are utilized as a reporter with different molecular combinations to label many biomolecular targets, including genomic and metabolic elements, even with multiplex targeting. Along with the advancements in barcoded bioassay, the improvements of various designs of barcode components, encoding and decoding strategies, and their portable adoption are indispensable in satisfying multiple purposes, such as medical confirmation and point-of-care (POC) testing. This perspective briefly discusses the current direction and progress of barcodes development and provides a hypothesis for barcoded bioassay in the near future.

## 1. Introduction

A barcode represents a collection of data which could be decoded by an optical readout [1]. Barcode technology was first innovated and adopted in industry. Due to the robust practicability, applications of barcode spread to many areas including clinical medicine [2], chemical detection [3], and retail trade [4]. Traditional barcodes deliver a variety of information by changing the width and gap of lines, which is identified as a linear or one-dimensional (1D) barcode. Instead of just a bar, new barcode developments involve geometric figures which are adapted to produce two dimensional (2D) recognition codes, such as quick response codes (QR code) [5]. Accompanying the change of code profiles, code readers are also undergoing innovation to achieve higher specification and sensibility. On the other hand, various label-free detection methods have been developed to avoid the labeling process, such as surface plasmon resonance [6], electrochemical detection [7], and grating-coupled interferometry [8]. However, all these techniques have their own limitations. For example, to detect the electrical signal change in bioassay, electrochemical impedance spectroscopy is always utilized [9,10]. It requires time-consuming calibration, and the sensitivity is poor without target amplification or signal amplification [11]. Thus, code labeling techniques, providing high sensitivity, selectivity, and multiplexing capability, are widely adopted in molecular diagnosis. 

Recent studies have demonstrated the feasibility of multiple barcodes constituted of biomolecules and particles for biological analysis. These barcodes could provide the function of molecular identification, which was proverbially proved valid in fundamental and laboratory analysis [12]. 

Various barcodes with high coding capability bring possibilities of multiplex detection. With the pursuit of higher specificity and sensitivity, barcodes might initiate a revolution for clinical detection. In this perspective, we provide a summary about the different types of barcodes, and predictions of future development and application of barcodes for health care, biological analysis, molecular detection, and in vivo diagnosis, as illustrated in Figure 1. Traditional barcodes are utilized for labeling the information and treatments of patients to prevent medication errors in health care [13]. Recently, new types of barcodes, also known as 2D recognition code, with or without fluorescence, are widely used to label target molecules such as proteins, DNA, and RNA. Furthermore, these barcodes are applied for detecting and monitoring the biological activity in vitro and in vivo for bioanalysis [14,15]. In the future, these advantages of different barcodes could be upgraded to achieve many different formations, such as different 3D patterns.

## 2. Classification of Barcodes for Bioassay

The common barcodes used for bioassay could be categorized into two main classes: Fluorescent and non-fluorescent-based barcodes. These barcodes could be either self-encoding, or adopted by other encoding approaches. For example, fluorescent-based barcodes could be combined with genetic and graphic encoding methods.

Fluorescent-based barcodes are broadly adopted to bioassay, providing a wide range of fluorescent colors, which is due to their specific emission spectra, diversity of combination, or difference in structures [12]. Fluorescence is a principal element for expressing the identity of biomolecular targets. Nowadays, several types of fluorescent barcodes with different materials are used for bioassay. Fluorescent dyes are generally applied as barcode materials to report the targets. Plenty of ready-made instruments used for excitation and emission detection have already extended the application of fluorescent dyes. Mixing different fluorescent dyes in various ratios is a useful strategy to create many different fluorescent colors, which could be individually or combinatorically applied as the tags to label the targets [16,17]. A string-like barcode employs sequencing dyes which are placed at a single-stranded backbone contained in the reporter probe [18]. This technology was utilized to provide genetic sequencing on an identified label via altering the order of fluorescent dyes to enhance specificity and reduce background (Figure 2).

According to our data, Ebola RNA molecules labeled by a string-like fluorescent barcode could be detected at both excitation of 532 nm and 633 nm (Figure 3) under a Total Internal Reflection Fluorescence (TIRF) microscope [19]. The diffusion of the labeled molecules was imaged with an electron multiplying CCD camera, enabling high-speed single molecule tracking. Because the barcode dye can be excited by different wavelengths, it improves the identification accuracy of target molecules by confirming the presence of targets in different channels. As shown in Figure 3, we are able to accurately identify the labeled Ebola RNA in dual channels with concentrations ranging from 300 fM to 30 pM. When the target concentration is low (300 fM), the target molecules are confirmed by both excitations (red boxes). Exploiting the sequence combination, changing the sequence of the fluorescent dyes on the backbone of the barcode dye can provide genomic information of different targets, resembling traditional barcodes [20], thus allowing highly multiplexing detection of the target molecules. For example, by changing the arrangement of four different colors, multiplexing detection of up to 800× can be achieved [18].

Fluorescent micro/nanostructures, such as quantum dots [21], lanthanide nanocrystals [22], and nanowires [23], have also been employed for bioassays. These sub-microstructures are facile and stable synthetic particles which provide changeable size, morphology, and different components with various surface modification available for different purposes. The probes could bind with these fluorescent sub-microstructures to label targets.

Recently, the common barcodes are integrated with several encoding methods, such as using DNA/RNA and immune antibodies encoding to report the targets with the aid of nanoparticles or nanorods, dyes, and others as reporters [24,25]. We used DNA oligonucleotide as a linker to bind the target by hybridizing with complementary oligonucleotide, and the accompanying fluorescent dye as a reporter to transfer the information of targets [20]. Similarly, gold nanoparticles as reporters could be detected not only by spectrometry because of their function of Raman signal enhancement, but also by naked eye because of changeable color by their aggregation and dispersion [26]. Furthermore, combining DNA oligonucleotide, antibody, and nanoparticles as a barcode was reported to achieve twice the signal amplification for protein detection [27,28]. These techniques contributed to establishing the non-fluorescent based barcode library (Figure 4). Structures could also be non-fluorescent based barcodes which need reporters labeling/detecting assistance, such as magnetically tunable structural colors, to help researchers to read out the information of targets in previous research [29]. These combinations give abundant possibilities for encoding and decoding the barcodes, and amplifying the barcode library.

## 3. Perspective on Barcodes

In previous research, barcodes were prevalently used for detecting proteins and nucleic acids in vitro [27,30]. Pre-synthesized DNA sequences worked as cellular barcodes to label and track large numbers of cells [31]. Along with the development of genome engineering technologies, in vivo barcodes are created using nuclease-induced mutations which extends barcode applications to establishing cell development and lineage tracing mammalian model systems [15,32]. With the in vivo barcodes, researchers could not only trace changes of the genome, but are also capable of marking and editing the mutated locus to reduce the incidence of lethal diseases, such as cancers. According to this information, preventive healthcare could be performed to potentially prevent the causes of potential diseases. This will dramatically improve the understanding of the mechanism of cause and development of disease and restoration of tissue and organs.

In bioassay, similar to the most commonly used detection technologies such as immunohistochemistry and immunofluorescence technologies, barcodes are designed as labels to tag and identify the known genomes [21] and proteins [33]. The purpose of barcodes application is to find and extract these targets out from complicated samples, such as tissues and bodily fluids [18]. These barcodes are mostly utilized for substance detection. However, in the past few years, barcodes reflecting the intracellular reactions, such as protein–protein and cell–cell interactions, were discussed by several studies. Schlecht et al. took two groups of random nucleotide barcodes and inserted them into different kinds of yeast cells to yield large libraries of double barcoding system [34]. Using this system, Schlecht and colleagues created a protein–protein interaction sequencing platform. Similar to protein–protein interactions, random mRNA barcodes were also used to target pre- and postsynaptic neurons to investigate the synaptic connections. In this study, the connectome protein was extracted after associating the neurons through protein–protein crosslinking across the synapse for sequencing [35]. Instead of reporting the information of substance, this research demonstrated the biological behavior based on the information from barcodes.

Indeed, bioreactions and behaviors are happening all the time in organisms. During the physiological and pathological reactions, energy could be generated, stored in the form of adenosine triphosphate (ATP), and transferred to support other reactions or produce heat. Tantama et al. used a biosensor to monitor the range of intracellular ATP:ADP in mammalian cells to observe the physiological changes in energy consumption and production [36]. In the near future, produced and consumed energy inside organisms could also be labeled and tracked to investigate the transportation pathways of energy. The barcodes could not only show the quantity of energy production and emission, but also report the creating or consuming ratio of energy by the luminous intensity.

The behaviors and reactions themselves were also considered as barcodes in previous studies, such as the intercellular protein phosphorylation [37,38,39,40]. Phosphorylation barcode has boundless prospects for detecting interaction of G proteins and G protein-coupled receptors (GPCRs) [37,39]. As the largest class of signal proteins in humans, the phosphorylation of GPCRs mediated by isoform-specific GPCR kinases leads to distinct phosphorylation patterns which produce conformational change of β-arrestins and downstream responses [37,39]. These serial reactions have substantially distinct phosphorylation patterns between even closely related GPCRs stimulated by the same stimulation. Therefore, it could be considered as a natural barcode through the mechanism of GPCRs phosphorylation. There are many molecule–molecule and cell–cell specific behaviors and interactions inside of an organism, and they might be responsible for the corresponding downstream reaction. These behaviors and interactions may be used as the barcodes in the future to distinguish their specific downstream reaction.

Currently, 2D recognition codes are used for most detections. In the future, 3D identification codes, known as shape recognition, will be created as the next generation in keeping with the times. These 3D identification codes could have various shapes, as well as 2D profiles. Instead of a planar surface, 3D codes could have uneven topography, which will provide adequate surface area for patterns with different reporters such as fluorescence, colors, and nanoparticles, and be detected by spectrometers and microscopes (Figure 5). With these advantages, 3D codes would possess striking labeling capability compared with 2D recognition codes. With these 3D identification codes, hundreds of thousands of options are possible by changing patterns and colors, bringing multiplexing detections which could be achieved easily and rapidly. It might also speed up the detections in testing organizations, and improve the quick response for diseases. State-of-the-art technology, such as 3D printing, both top-down [41] and bottom-up [42], could provide candidates for these 3D codes fabrications. Furthermore, they might be directly created from 2D to 3D using origami technology [43]. With the development of barcodes, quantifiable data of 3D codes could be exported by a particular reader and analyzed by simple and convenient methods.

Barcodes are robust tools and are applied to various fields for reporting information on different targets. To investigate the physiological and pathological phenomenon and the mechanisms of these targets, barcodes are prevalently utilized for bioassay. By means of barcodes, from the microscale molecules to macroscale cells, one signal phenomenon to serial reactions could be distinguishable. In the future, diversified forms of barcodes could be created and adopted for different purposes in the foreseeable future along with the development of scanning and readout devices, which could launch a new revolution for exploring the occurrence and development of physiological and pathological processes.

## Figures and Tables

**Figure 1 diagnostics-09-00075-f001:**
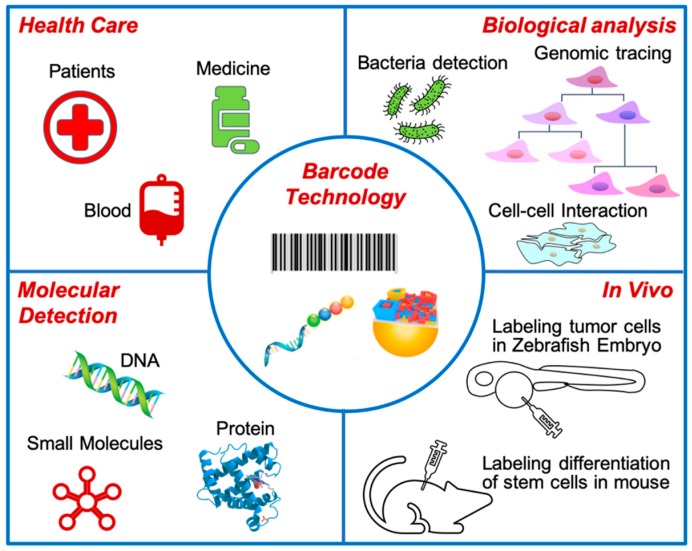
Application of barcodes. Barcodes are prevalently applied to a bunch of medical and biological fields.

**Figure 2 diagnostics-09-00075-f002:**
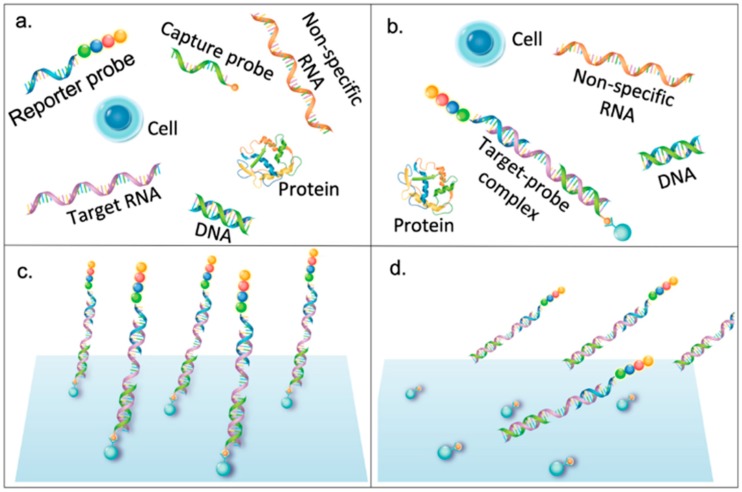
A string-like barcode dye for target RNA labeling and detection. (**a**) Capture probe and reporter probe are mixed with biological samples containing target RNA. (**b**) Capture probe and reporter probe bind with the target RNA by complementarily binding with target RNA using their short nucleotide fragment. (**c**) After immobilizing the complex on the solid surface, the residual is washed away. (**d**) The complex of capture probe, reporter probe, and target RNA is released by ultraviolet light exposure.

**Figure 3 diagnostics-09-00075-f003:**
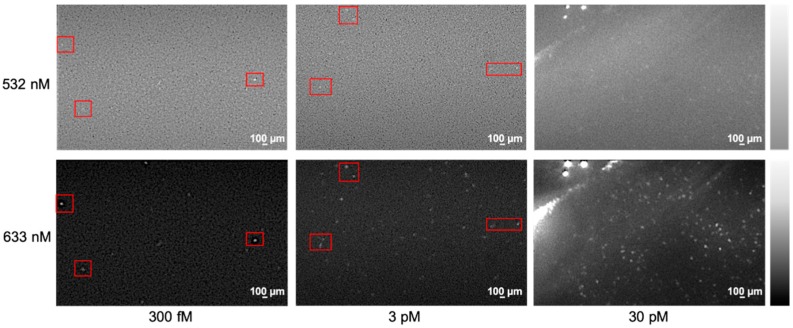
A string-like fluorescent barcode dye-labeled RNA target with various concentrations (300 fM–30 pM) imaged by a Total Internal Reflection Fluorescence (TIRF) microscope with dual channel excitation (532 nm and 633 nm). The red boxes show the molecules being observed unber both excitations.

**Figure 4 diagnostics-09-00075-f004:**
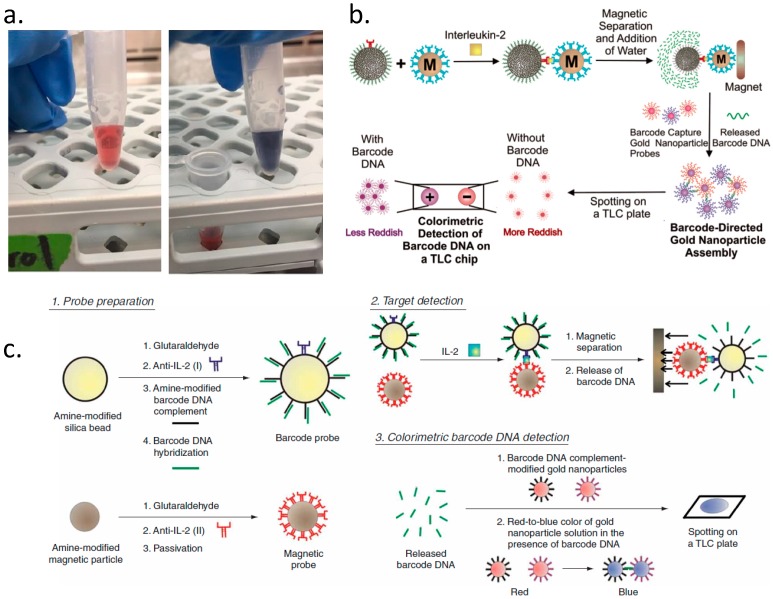
Non-fluorescent barcodes. The colorimetry detection is realized by DNA hybridization based gold nanoparticle aggregation. (**a**) The separated gold nanoparticle (red) are aggregated (blue) after DNA hybridization, reprinted with permission from Reference [26] Copyright (2017) biosensors. (**b**) The gold nanoparticle related colorimetry could achieve attomolar (aM) level sensitivity by combining DNA oligonucleotide with antibody and particles, reprinted with permission from Reference [27] Copyright (2005) American Chemical Society. (**c**) With similar technology, the gradient concentration of targets could be detected according to the color change of gold nanoparticle reagent, reprinted with permission from Reference [28] Copyright (2007) Springer Nature.

**Figure 5 diagnostics-09-00075-f005:**
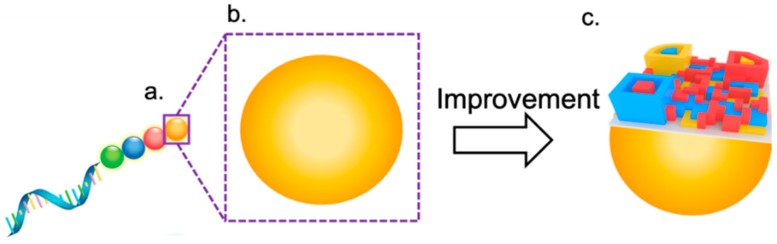
Schematic of future 3D identification code for biosensing with various and distinguished 3D patterns of colors, shapes, heights, and widths.

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
