# Peer review of "Perspective of Molecular Diagnosis in Healthcare: From Barcode to Pattern Recognition"

_diagnostics, 2019, doi:10.3390/diagnostics9030075_

Round 1
Reviewer 1 Report
I am unhappy about this paper:
On the one hand I acknowledge the review of various methodologies to be influenced by “barcodes”, which certainly will be useful for many readers in the whole field of molecular biology and biochemistry.
However, on the other hand I have serious doubts, whether the term “barcode” should be extended in this way: “Barcode” includes “bar”, i.e. a linear (1D) structure, not a 2D or a 3D structure/pattern. For 2D-recognition the term “QR-Code” is well established, 3D-identification runs under “shape recognition” or “matlab/python code”.
Indeed, I do not see any advantage to summarize all these codes resp. recognition modes under “barcode”, but instead would still prefer “pattern recognition data” for all of them.
So, as far as linear structures/patterns are meant, the term “barcode” is OK for various applications, but don´t extend the term to 2D or 3D-structures/patterns.
Author Response
Dear Editor:
Thank you for the opportunity to respond to the Reviewers’ comments. We address each
comment in the attached file.
We appreciate the opportunity to have our manuscript considered for publication in MDPI Diagnostics.
Regards,
Ke Du
Reviewer 1:
One the one hand I acknowledge the review of various methodologies to be influenced by “barcodes”, which certainly will be useful for many readers in the whole field of molecular biology and biochemistry.
However, whether the term “barcode” should be extended in this way: “Barcode” includes “bar”, i.e. a linear (1D) structure, not a 2D or a 3D structure/pattern. The advantage is not clear to summarize all these codes resp. recognition modes under “barcode”, as far as linear structures/patterns are meant, the term “barcode” is OK for various application, but don’t extend the term to 2D or 3D-structures/pattern.
Response:
We appreciate your comments. According to your suggestion, we replaced the words of 2D barcode and 3D barcode by 2D recognition code and 3D identification code. The details of modification as below:
On Page 1, Line 32, we changed 2D barcode to 2D recognition codes.
On Page 5, Line 204 to 206, we changed 2D barcode to 2D recognition codes and 3D barcode to 3D identification codes, respectively. Line 206, 3D barcodes is replaced by 3D codes. Line 208, 3D barcodes was modified to 3D codes. Line 209, 2D barcode was modified to 2D recognition code, and 3D barcode is changed to 3D identification code.
On Page 6, Line 273, 3D barcodes was changed to 3D codes, and Line 278, we changed 3D barcode to 3D identification code.
One Page 9, Line 420 and Line 422, we changed 2D barcode to 2D recognition code and 3D barcode to 3D identification code (shape recognition).
The description of each code in the figure of Abstract graphic is also changed according to your suggestion.
Reviewer 2 Report
The manuscript provides a perspective on the relevance of barcode technology in different fields, including a discussion of progress of barcodes development. The perspective is well writing and scientifically y sound. The subject is of relevance for different fields. I recommend the following changes.
1) It would be useful to see a table summarizing the main applications of barcodes, including the most relevant references.
2) The manuscript should also include a brief comparison with alternative approaches in terms of outcomes, time do implement, costs, etc. The authors can choose one application as an example.
3) The “hypothesis for barcoded bioassay in the near future” described in the Abstract is not clear in the main text. Please clarify this issue in the main text.
4) Figure 2 seems too specific for a broad perspective. The authors should considered to use images that convey the features of barcodes in a more clear way to the reader.
Author Response
Dear Editor:
Thank you for the opportunity to respond to the Reviewers’ comments. We address each
comment below.
We appreciate the opportunity to have our manuscript considered for publication in MDPI Diagnostics.
Regards,
Ke Du
Reviewer 2:
The manuscript provides a perspective on the relevance of barcode technology in different fields, including a discussion of progress of barcodes development. The perspective is well writing and scientifically sound. The subject is of relevance for different fields. I recommend the following changes.1) It would be useful to see a table summarizing the main applications of barcodes, including the most relevant references.2) The manuscript should also include a brief comparison with alternative approaches in terms of outcomes, time do implement, cost, etc. The authors can choose one application as an example.3) The “hypothesis for barcoded bioassay in the near future” described in the Abstract is not clear in the main text. Please clarify this issue in the main text.
4) Figure 2 seems too specific for a broad perspective. The authors should considered to use images that convey the features of barcodes in a more clear way to the reader.
Response:
We appreciate your comments. According to your suggestion, we revised the content of our manuscript as below:
1) In order to summarize the application of barcodes which used for many fields, we inserted a chart in the main content of manuscript on page 2 between Line 62 and Line 63, and added description of Chart 1 from line 55 to 62 including more references (reference 13 to 15):
“…application of barcodes for health care, biological analysis, molecular detection, and in vivo diagnosis, as illustrated in Chart 1. Traditional barcodes are utilized for labeling the information and treatments of patients to prevent medication errors in health care [13]. Recently, new types of barcodes, also known as 2D recognition code, with or without fluorescence are widely used to label target molecules such as proteins, DNA and RNA. Furthermore, these barcodes are applied for detecting and monitoring the biological activity in vitro and in vivo for bioanalysis [14, 15]. In the future, these advantages of different barcodes could be upgraded to achieve many different formations, such as different 3D patterns.”
The legend of Chart 1 was also added on Line 66:
“Chart 1. Application of Barcodes. Barcodes are prevalently applied to a bunch of medical and biological fields”
2) According your suggestion, we added the description of barcode technology comparing with label free technology on page 1, Line 34 to Line 40, including more references of different label free technologies (reference 6 to 8) and their limitation (reference 9 to 11):
“On the other hand, various label-free detection methods have been developed to avoid the labeling process, such as surface plasmon resonance [6], electrochemical detection [7] , and grating-coupled interferometry [8] . However, all these techniques have their own limitations. For example, to detect the electrical signal change in bioassay, electrochemical impedance spectroscopy is always utilized [9, 10]. It requires time-consuming calibration and the sensitivity is poor without target amplification or signal amplification [11]. Thus, code labeling technique, providing high sensitivity, selectivity, and multiplexing capability is widely adopted in molecular diagnosis.”
3) The hypothesis of 3D codes are added and modified on Page 5, Line 203 to Line 212 according to the revision:
“Currently, 2D recognition codes are used for most detections. In the future, 3D identification codes, known as shape recognition, will be created as the next generation in keeping with the times. These 3D identification codes could have various shapes as well as 2D profiles. Instead of a planar surface, 3D codes could have uneven topography, which will provide adequate surface area for patterns with different reporters such as fluorescence, colors, and nanoparticles, and be detected by spectrometers and microscopes (Figure 4). With these advantages, 3D codes would possess striking labeling capability compared with 2D recognition codes. With these 3D identification codes, hundreds of thousand of options are possible by changing patterns and colors, bringing multiplexing detections which could be achieved easily and rapidly. It might also speed up the detections in testing organizations and improve the quick response for diseases.”
4) The results we put in Figure 2 are unique data showing the high sensitivity and multiplexing capability of the barcode dyes. Based on our knowledge, these results were never published before. It demonstrates that barcode technology can label the target nucleic acid and imaged with different wavelengths. We added more experimental details of our results on Page 3, Line 96 to Line 106 to help the readers better understand our results:
“ According to our data, Ebola RNA molecules labeled by a string-like fluorescent barcode could be detected at both excitation of 532 nm and 633 nm (Figure 2) under a total internal reflection fluorescence (TIRF) microscope [19]. The diffusion of the labeled molecules was imaged with an electron multiplying CCD camera, enabling high speed single molecule tracking. Because the barcode dye can be excited by different wavelengths, it improves the identification accuracy of target molecules by confirming the presence of targets in different channels. As shown in Figure 2, we are able to accurately identify the labeled Ebola RNA in dual channels with concentrations ranging from 300 fM to 30 pM. When the target concentration is low (300 fM), the target molecules are confirmed by both excitations (red boxes). Exploiting the sequence combination, changing the sequence of the fluorescent dyes on the backbone of the barcode dye can provide genomic information of different targets, resembling traditional barcodes [20], thus allowing highly multiplexing detection of the target molecules. For example, by changing the arrangement of four different colors, multiplexing detection of up to 800´ can be achieved [18].”
In order to clearly transmit concept of barcode, we changed the title of this manuscript to “Roadmap of Molecular Diagnosis: Barcode, Quick Response Code, and Pattern Recognition”.
Additional modifications are described at below in detail:
1. Page 1, Line 30, “…gap of lines which is …” was modified to “…gap of lines, which is …”, and “…one-dimensional…” to “…one-dimensional (1D)…”
2. Page 1, Line 43, “identification which was…” was modified to “identification, which was…”
3. Page 2, Line 70, “fluorescence colors which is…” was modified to “fluorescence colors, which is…”
4. Page 2, Line 71 to Line 72, “…identity of targets…” was modified to “…identity of biomolecular targets…”
5. Page 3, Line 90, “…target RNA by complimented with…” was modified to “…target RNA by complimentarily binding with…”
6. Page 3, Line 111, “These submicro-structures…” was modified to “These sub-microstructures…”
7. Page 4, Line 120, “…these submicro-structures…” was modified to “…these sub-microstructures…”
8. Page 4, Line 129, “aggregation and separation.” was modified to “aggregation and dispersion.”
9. Page 4, Line 130, “…nanoparticles as a barcode was…” was modified to “…nanoparticles as a barcode were…”
10. Page 4, Line 151, “…model system [15, 32].” was modified to “…model systems [15, 32].”
11. Page 4, Line 152 to 153, “…, but also capable to” was modified to “…but are also capable of”
12. Page 5, Line 164, “mark and edit the mutated locus…” was modified to “marking and editing the mutated locus…”
13. Page 5, Line 165, “…to relatively prevent…” was modified to “…to potentially prevent…”
14. Page 5, Line 171, “… from complex samples, …” was modified to “… from complicated samples, …”
15. Page 5, Line177, “interactions, in previous study tried to use random mRNA barcodes…” was modified to “…interactions, random mRNA barcodes were also used…”
16. Page 6, Line 273, “…, both top down and bottom up…” was modified to “…, both top-down and bottom-up…”
17. Page 6, Line 276, “…be exported by a special…” to “…be exported by a particular…”
18. Page 6, Line 280, “…information of different…” to “…information on different…”